# Preoperative Fasting and General Anaesthesia Alter the Plasma Proteome

**DOI:** 10.3390/cancers12092439

**Published:** 2020-08-27

**Authors:** Ulf Gyllensten, Sofia Bosdotter Enroth, Karin Stålberg, Karin Sundfeldt, Stefan Enroth

**Affiliations:** 1Department of Immunology, Genetics, and Pathology, Biomedical Center, Science for Life Laboratory (SciLifeLab) Uppsala, Box 815, Uppsala University, SE-751 08 Uppsala, Sweden; ulf.gyllensten@igp.uu.se; 2Medical Products Agency, Box 26, SE-751 03 Uppsala, Sweden; sofia.bosdotter.enroth@mpa.se; 3Department of Women’s and Children’s Health, Uppsala University, SE-751 85 Uppsala, Sweden; karin.stalberg@kbh.uu.se; 4Department of Obstetrics and Gynaecology, Institute of Clinical Sciences, Sahlgrenska Academy at Gothenburg University, SE-416 85 Gothenburg, Sweden; karin.sundfeldt@obgyn.gu.se

**Keywords:** plasma proteome, preoperative conditions, ovarian cancer, anaesthesia, short-time fasting

## Abstract

Background: Blood plasma collected at time of surgery is an excellent source of patient material for investigations into disease aetiology and for the discovery of novel biomarkers. Previous studies on limited sets of proteins and patients have indicated that pre-operative fasting and anaesthesia can affect protein levels, but this has not been investigated on a larger scale. These effects could produce erroneous results in case-control studies if samples are not carefully matched. Methods: The proximity extension assay (PEA) was used to characterize 983 unique proteins in a total of 327 patients diagnosed with ovarian cancer and 50 age-matched healthy women. The samples were collected either at time of initial diagnosis or before surgery under general anaesthesia. Results: 421 of the investigated proteins (42.8%) showed statistically significant differences in plasma abundance levels comparing samples collected at time of diagnosis or just before surgery under anaesthesia. Conclusions: The abundance levels of the plasma proteome in samples collected before incision, i.e., after short-time fasting and under general anaesthesia differs greatly from levels in samples from awake patients. This emphasizes the need for careful matching of the pre-analytical conditions of samples collected from controls to cases at time of surgery in the discovery as well as clinical use of protein biomarkers.

## 1. Introduction

Biological biomarkers for disease can be used for screening, differential diagnostics, non-invasive detection of malignancies, progress of a particular disease or aid in selection of treatment. The ideal biomarker should display deviating levels between healthy individuals and those with the studied outcome and preferably not be affected by non-disease related factors such as anthropometrics, hormonal status, age, ethnicity, medication, or individual lifestyle choices including smoking or dietary habits. Previous studies have shown that the majority of plasma or serum protein biomarkers are affected by a wide range of factors which can affect their applicability, sensitivity and specificity in relation to a particular disorder [1,2,3,4,5,6]. Detection of biomarkers for a disease is often performed using samples collected over time, which adds additional sources of variation such as time of storage in the freezer or season [7]. Pre-analytical variables [8] such as storage time, thaw-freeze cycles, equipment used in collection and analysis could also affect the protein levels and such conditions needs to be normalized between cases and controls to ensure that the effect of the disease is studied [5]. Fresh-frozen tissues collected at time of clinical surgery is an excellent source of material for investigations into disease aetiology and for discovery of novel biomarkers. In addition to primary tissue, blood samples are often obtained that allows for investigation into, for instance, genetic differences between tumour DNA and the normal genetic constitution. Such samples are in general collected in relation to planned surgery, which ensures adequate handling of samples and minimized time between the samples are taken and downstream processing, freezing and storage. Before surgery, patients are recommended over-night fasting in order to reduce potentially life-threatening risks during general anaesthesia, such as pulmonary aspiration. Such short-term fasting has been shown to effect the human metabolism [9] for instance reducing the insulin-like growth factors I (IGF-I) and increasing IGF binding protein-I (IGFBP-I) in sera [10]. Several of these effects are related to glucose production through glycogen breakdown entering the blood stream from the liver when substrate levels are low [9], especially in the early stages of fasting [11]. We have previously reported that several types of the most common medications have large effects on the human plasma proteome [3], and it is therefore possible that drugs administered in direct relation to planned surgery such as tranquilizers and antibiotics could also affect the circulating levels of plasma proteins. In addition to preoperative conditions, a previous study has investigated the effects of general anaesthesia on three protein biomarkers (Squamous Cell Carcinoma antigen (also known as ‘Serpin B3’), Carcinoembrionic Antigen (CEA) and CYFRA 21-1 (a soluble fragment of ‘Keratin, type I cytoskeletal 19′, KRT19)) in plasma from patients with lung cancer. In that study, these biomarkers display significantly deviating levels between samples collected before and after anaesthesia [12]. Another study by Thorpe and colleagues investigated three protein biomarkers (Mucin-16, also known as CA-125), Prolactin (PRL) and Macrophage Migration Inhibitory Factor (MIF) in healthy women and women undergoing gynaecologic surgery in relation to ovarian cancer [13]. Mucin-16 was introduced as a biomarker for ovarian cancer in 1983 [14] and is currently the most important single biomarker [15] with increasing circulating levels as the cancer progresses [16]. Thorpe and colleagues [13] found strong effects by general anaesthesia on circulating Prolactin levels, replicating a previous result for the same protein [17] but found no statistical significant effects on Mucin-16. Hagen et al. [17] showed that while Prolactin levels increased under general anaesthesia, they did not for patients under epidural analgesia. These studies indicate that pre-analytical conditions need to be carefully considered in biomarker discovery studies, in particular if the cases are collected at time of treatment, during surgery and under general anaesthesia.

We performed a comprehensive study of the effects of general anaesthesia compared to being awake at sample collection on the plasma proteome profile. We evaluated a total of 983 unique plasma proteins in three sets of patients diagnosed with ovarian cancer. The first set of samples was collected at time of diagnosis from awake patients, while the second and third sets of samples were from patients undergoing planned surgery after general anaesthesia. No patients overlapped between the cohorts. Our two-stage study design replicate the effect of anaesthesia on 131 proteins. A combined analysis further indicates that 421 of the proteins display significantly deviating levels between the two pre-analytic conditions. Finally, we demonstrate how these effects can yield false positive results by comparing the plasma protein levels in the cases to awake controls with implications in cancer-related pathways.

## 2. Results

### 2.1. Effects of Sampling at Surgery Compared at Time of Diagnosis

We first performed a univariate comparison of the z-transformed protein values (Methods) measured in the Swedish U-CAN cohort with samples collected from awake patients at time of diagnosis and the 1st cohort of samples collected at surgery. In the discovery cohort (Table 1), a total of 593 assays were performed, corresponding to 586 unique proteins (uniprot IDs). Using a Bonferroni adjusted cut-off (*p* < 8.43 × 10^−5^ = 0.05/593), 155 assays corresponding to 153 unique proteins were found to have significantly deviating levels between the two pre-analytic conditions (Appendix A). Out of the proteins characterized in duplicates across multiple Olink panels, all replicates were either non-significant or significant, and the two that were significant had similar effect sizes (Appendix A). 154 of these 155 proteins were also measured in the replication cohort and among these 154 proteins, 92.2% (142/154) were nominally significant also in the replication cohort (Table 1) and 85.1% (131/154) remained significant after multiple hypothesis correction (Appendix A). Of these 131 proteins, 2 (Aldehyde Dehydrogenase 3 Family Member A1 (ALDH3A1) and Leukotriene A4 hydrolase (LTA4H)) were found to have higher levels in the samples that were collected at surgery compared to samples taken at time of diagnosis, and 129 were found to have lower levels (Appendix A). We found 11 (SERPINA7 (Thyroxine-binding globulin), ROBO1 (Roundabout homolog 1), CDH1 (Cadherin-1), CD209 (CD209 antigen), ST6GAL1 (Beta-galactoside alpha-2,6-sialyltransferase 1), LRP11 (Low-density lipoprotein receptor-related protein 11), CDON (Cell adhesion molecule-related/down-regulated by oncogenes), VASH1 (Vasohibin-1), IL13RA1 (Interleukin-13 receptor subunit alpha-1), CEACAM1 (Carcinoembryonic antigen-related cell adhesion molecule 1) and SPINK5 (Serine protease inhibitor Kazal-type 5)) of the 129 proteins with lower abundance levels in samples taken at surgery to also have mouse orthologs shown to be repressed as a result of 24 h fasting [11]. This indicates that the reduced protein levels we observe may reflect both fasting and anaesthesia, however these effects cannot be separated in our study.

Among the proteins from multiplex panel 6–11 (Table 1), 86.8% of the proteins replicated while only 60.0% of the proteins from panel 1–5 (Table 1) remained significant in the replication cohort. All cohorts in our study are powered to detect differences of 0.50–0.58 (z-scale) at 95% power and a significance level 0.05, except the replication of panel 1–5 which is powered to detect differences of 0.76. The mean of the absolute differences in the discovery cohort among the proteins replicated was 0.73 and in the non-replicated 0.71, suggesting that the replication cohort for panel 1–5 is underpowered. We then investigated if any particular class of function or pathway was enriched among the proteins that were affected. Since the selection of proteins present on the available PEA panels does not represent a random selection, we therefore performed Gene Ontology (GO) and KEGG pathway enrichment analysis using the online DAVID-tool [19,20] with the complete list of assays used here as background and the list of replicated protein changes as foreground. Using this setup, we found a statistically significant enrichment of “cell adhesion” (GO: 0007155, *q*-value = 0.023, adjusted for multiple hypothesis testing by Bonferroni correction). There was no further statistical enrichment of GO-terms nor any KEGG-pathways.

### 2.2. Combined Analysis and Replication of Previously Published Results

We then performed a combined analysis using all available samples and proteins. In this analysis, 437 assays corresponding to 421 unique proteins were found to have significantly deviating levels between the two pre-analytic conditions (Appendix A, Figure 1a). The protein with the largest change was Prolactin with a mean normalized protein expression (NPX, Methods) level of 12.0 in sedated patients and 10.0 in awake patients (Figure 1b). Since the NPX is on a log2-scale, this corresponds to 4 times higher values and is comparable to the levels observed by Hagen [17] with 10–30 μg/L at base level and 100–120 μg/L at 0.5 to 2 h after general aesthesia.

Thorpe and colleagues [13] reported statistically significant changes in serum Prolactin levels in the same direction as Hagen’s [17] plasma concentrations in relation to anaesthesia. Thorpe also analysed Mucin-16 and MIF but found no significant effects. Here, nominally significant changes were found for Mucin-16 (+0.90 NPX in sedated patients, *p* = 0.02, Bonferroni adjusted *q* = 1.0) and MIF (−0.66 NPX in sedated patients, *p* = 5.73 × 10^−5^, *q* = 0.06), but none remained significant after adjustment for multiple hypothesis testing.

Two of the three proteins analysed in sera by Kahn et al. [12] were also analysed here, although in plasma. In Kahn’s study, CEA was reported at surgery to have 0.70 times mean concentration (*p* = 0.0039) compared to before surgery, and CYRFA 21-1 to have 1.51 times mean concentration (*p* = 0.0479). In our study, all observed differences were in the same direction as in Kahn’s study, but the changes in plasma levels of CEA were not significant (−0.30 NPX, *p* = 0.31), and similarly for KRT19 although the change was nominally significant (+0.69 NPX, *p* = 0.012) it was not after adjustment for multiple hypothesis testing (Bonferroni adjusted *q* = 1.0). As above, we performed Gene Ontology (GO) and KEGG pathway enrichment analysis using the online DAVID-tool [19,20] with the complete list of assays used here as background and the list of significantly deviating proteins as foreground. Using this analysis, we found no statistical enrichment of GO-terms (molecular function or biological process) nor KEGG-pathways (*q*-values > 0.05, adjusted for multiple hypothesis testing by Bonferroni correction).

Even though the genes encoding proteins with significantly deviating levels between samples collected at time of diagnosis in wake patients and samples collected at time of surgery were not statistically overrepresented in specific pathways, genes encoding proteins with such changes were represented in each of the 25 cancer pathways listed in the KEGG database (Figure 2a, Appendix A). As an example, we plotted out the first two components of PCA (principle component analysis) of untransformed NPX values for proteins for two such pathways. As can be seen from Figure 2b, in proteins with encoding genes in the “Central carbon metabolism in cancer” pathway, this leads to clustering of samples taken at time of surgery (orange and red) and healthy controls (black), while the majority of samples collected at time of diagnosis (blue) forms a distinct separate cluster. For other pathways, such as “Choline metabolism in cancer” there is no clear pattern separating the cases and controls (Figure 2c). This indicates that without proper matching of cases and control samples, both false positive and false negative results can easily be obtained in analysis of plasma protein biomarkers.

### 2.3. Stage-Independent Effects by Sedation on Mucin-16 for Ovarian Cancer

Finally, we compared clinical pre-operative Mucin-16 levels for two sample sets both collected from awake patients but from two separate locations. Measurements were available for 75 of the 77 samples from the U-CAN biobank, and from a set of samples (*n* = 88) from Sahlgrenska academy, Göteborgs Universitet. The samples from Göteborg used here are not from the same women as the cohorts used in the main analysis of effects on sedation above and no additional protein measurements were available from this cohort. There was however a significant correlation between the clinical measures for the U-CAN samples and the PEA measurements (spearman’s rho = 0.56, *p* = 4.4 × 10^−5^). We found no statistical difference (Figure 3a) in the clinical measurements between the two sites, neither when comparing all stages together (*p* = 0.96, two-sided wilcoxon-test) or when comparing stages separately (all *p* > 0.55, two-sided wilcoxon-test). We then modelled the observed Mucin-16 values as a function of stage (Figure 3b) for the two sites and although the inclination of the models is different, it is clear that both measurements increase with the progression of cancer as indicated by stage. We then performed a similar analysis but for the protein abundance levels as measured by PEA in the sedated and wake cohorts (Figure 3c,d) and found similar inclination of the models for sedated and wake samples but with sedated having consistently higher levels. As a comparison, the levels of Prolactin (Figure 3e,f) did not vary between ovarian cancer stages but had clearly separated distributions in sedated and awake samples.

## 3. Discussion

Differences in pre-analytical factors can introduce bias in biomarker discovery studies [8]. Previous studies have, for instance, shown that time to centrifugation but not number of freeze-thaw cycles commonly affect plasma protein levels [21]. We have previously shown that long-time freeze storage influences measurable protein levels with similar proportion of explained variance as individual age [7]. In order to facilitate collection of samples that have undergone similar handling, annotation standards such as the SPREC-coding have been developed [22] to standardize annotation of pre-analytical factors that could influence downstream analysis. In addition to sample preparation and handling, individual differences in lifestyle and genetics have been shown to explain up to 88% of the inter-individual variation observed in protein plasma biomarkers [1]. A few previous studies [12,13,17] have shown that commonly used anaesthesia have large effects on protein biomarkers in sera or plasma. The study conducted by Hagen and colleagues in 1980 [17] showed that the effects on Prolactin levels differed between female patients under general anaesthesia as compared to epidural analgesia. This suggests different biological mechanisms and shows the importance of knowing the details of the sampling conditions. This has been further illustrated by the work of Kahn et al. [12] where both the effects of anaesthesia and venous versus arterial vessel puncture were investigated on three protein biomarkers in sera. Overall however, studies investigating the effects of anaesthesia on circulating protein levels are scarce and based on small sample sizes and few proteins. Kahn et al. [12] analysed 10 men and three women diagnosed with either non-pulmonary primary cancer (*n* = 3), non-small cell lung cancer (*n* = 9) or hamartoma of the lung (*n* = 1), and samples were collected at different time-points before and during anaesthesia. Thorpe [13] studied 3 proteins in sera collected from women diagnosed with ovarian cancer before surgery (*n* = 19) and at surgery (*n* = 46). Hagen’s study [17] encompassed 13 patients out of which 6 underwent epidural analgesia and 7 general anaesthesia with blood samples taken 12 times for each patient before, during and after admission and surgery. Here we have analysed up to 983 plasma proteins in up to 327 women diagnosed with ovarian cancer, with samples collected at diagnosis or at surgery after general anaesthesia but before incision. We replicated the results from two previous studies but did not replicate the results presented in Kahn’s study. The direction of change in protein levels here were the same as in Kahn’s study but did not reach statistical significance in our data. This can be due to several reasons; plasma as compared to sera, women only compared to men and women jointly and finally, patients with ovarian cancer compared to patients with lung cancer.

Our study is limited by several factors. Although there are no differences in age-distribution between the cohorts (Table 1), the fraction of stages according to the FIGO-standard (Table 1) varied significantly (Appendix A) for around half of the available stage-fractions between any two patient cohorts used here. Neither did we have access to detailed records of co-morbidity, menopausal stage, circadian rhythm or lifestyle choices such as smoking status or alcohol use. However, we analysed all stages and ages jointly and in our largest material, including the data for protein panels 6–11 (Table 1), over 85% of the associations found in the discovery step did replicate, suggesting that the results are robust despite the actual differences in stage-distribution and possible other differences between the cohorts. We also only have one sample from each woman, and this means that we cannot separate the effects of e.g., short-time preoperative fasting and the effects of anaesthesia. Measurable levels of plasma protein biomarkers are also known be affected by a wide range of factors, in addition to pre-analytical conditions, also anthropometrics and lifestyle variables. Here, samples were collected from two separate locations that employs similar sample collection protocols but with some differences. One such difference is the temperature at which the samples are stored between collection and the isolation of plasma and freezing. The samples collected from awake patients at time of diagnosis were stored at room temperature while the samples from sedated patients were stored at 4 °C. We found 135 proteins that overlapped between our study and a study [21] investigating the effects of storing samples at 4 or 22 °C for 1 or 3 h prior to plasma isolation and among these, up to 7 proteins were found by Shen and colleagues [21] to have deviating observable protein levels comparing 1 h to 3 h storage at 4 °C, 1 h storage at 22 °C vs. 4 °C or 3 h storage at 22 °C vs. 1 h at 4 °C. Among these 7 proteins, no significant enrichment was however found in the group of proteins identified here to be affected by sedation (*p* > 0.22, two-sided binomial test). As for anthropometrics and lifestyle variables the vast majority of these traits were either not recorded at the participating hospitals from where the samples in this study were obtained or could not be made available to us under the current ethical permits and so their direct effects cannot be investigated. In order to estimate if there were an influence by covariates on the proteins detected as affected or not by sedation, we downloaded a list of known covariates from a study characterizing five of the PEA-panels used here in a cross-sectional cohort with 159 covariates including sex, anthropometrics, lifestyle variables and use of common medications [3]. Among the proteins on the five overlapping PEA-panels we found no statistical difference in the fraction of affected proteins overall and in the sedation-affected group, neither for the nominally significant (*p* = 0.15, two-sided binominal test) nor when adjusted for multiple hypothesis testing (*p* = 0.94, two-sided binomial test) (Appendix A). Although circumstantial, this is indicative of that it is unlikely that any unrecorded general differences in e.g., lifestyle between the cohorts are the main determinants of the changes we observed in relation to sedation. In addition, as with the previous studies, we focused on patients undergoing surgical treatment for a specific disease. Studying the effects of anaesthesia in healthy subjects would however be unethical as anaesthesia is not completely risk free [23]. This could have been overcome by taking additional blood samples from the patients at time of surgery, before anaesthesia, but was not done here. Lastly, patients undergoing surgery was administered colloid fluids as replacement for loss of blood and as prevention of thromboembolic complications. Although we observe a reduction in protein abundance levels among sedated individuals compared to awake for several proteins in the coagulation cascade such as PROC (Vitamin K-dependent protein C, −0.70 NPX, *p* = 1.22 × 10^−25^), F7 (Coagulation factor VII, −0.59 NPX, *p* = 6.87 × 10^−24^) and F11 (Coagulation factor XI, −0.55 NPX, *p* = 2.38 × 10^−22^) (Appendix A), it cannot be ruled out that some of these reductions in protein levels could be due to dilution of the blood volume by the administered fluids. However, a recent study [24] investigating the effect on blood volume expansion by four different colloid fluids, including Macrodex^®^, estimated the initial expansion by volume to approximately 14%. Although speculative, a 14% expansion in blood volume and plasma would correspond to an observed difference of −0.13 NPX in a given protein concentration. All of the proteins with significant *p*-values after multiple hypothesis correction, differ more than 0.13 NPX between sedated and awake patients, and 96.6% of the significant changes have differences larger than two times that (0.26 NPX), suggesting that the differences in protein levels are not likely only due to a dilution of the plasma in the sedated individuals compared to the wake.

Many severe malignant diseases such as cancers of internal organs e.g., hepatocellular cancers or pancreatic cancers have few prognostic biomarkers and high mortality rates. Pancreatic cancers have only 6% 5-year survival rate and are often treated by major surgical resection [25]. Pancreatic cancers have a reported incidence of 1.0–7.4 per 100,000 individuals across the globe [26], but is increasing and there is an urgent need for non-invasive predictive early biomarkers. Several recent studies have studied circulating DNA as markers of carcinogenesis or biomarkers for early detection of pancreatic cancer, and in such studies it is not uncommon that the case samples are collected on the day of surgery but prior to any incision being made [27,28,29,30], while the controls are healthy volunteers. This implies that the patients have undergone at least overnight fasting while the controls have not. Although this does not likely introduce any bias in analyses of DNA, it could very well introduce large biases for epigenetics, transcriptomics and proteomics. The bio-banked plasma or serum samples originally collected for DNA analysis are likely to be studied using novel technologies for e.g., high-throughput ultra-sensitive RNA or proteomics [25] assays in the search of non-invasive circulating biomarkers. Of the proteins analysed here that showed significantly different levels in the combined analysis, 5 overlap with the KEGG “pancreatic cancer” pathway (NF-kappa-B essential modulator (NEMO), Pro-epidermal growth factor (EGF), Cyclin-dependent kinase inhibitor 1 (DKN1A), Vascular endothelial growth factor A (VEGF-A) and Transforming growth factor beta-1 (TGF-beta-1)). In addition, an additional 8 overlapping proteins have been suggested as potential biomarkers for pancreatic cancer (Hepatocyte growth factor (HGF), Interleukin 7 (IL-7), Cathepsin D (CTSD), Monocyte chemoattractant protein 1 (MCP-1), Interleukin-16 (IL-16), C-C motif chemokine 11 (CCL11), Osteoprotegerin (OPG) and Anterior gradient protein 2 homolog (AGR2)) [31,32,33,34,35]. Of these, all but AGR2 show lower levels (−0.59 to −1.53 NPX) in sedated compared to wake patients, while wake individuals have higher levels of AGR2 (+0.71 NPX) compared to sedated. This demonstrates that it will be of great importance to be aware of possible bias introduced by pre-analytical differences such as those described here, that could otherwise influence the results obtained.

## 4. Materials and Methods

### 4.1. Samples

The three samples sets used were from the U-CAN collection [36] at Uppsala Biobank, Sweden, and from Sahlgrenska Gyncancer Biobank, Göteborg, Sweden and have been described earlier [18,37]. The EDTA plasma samples from the U-CAN biobank were from patients diagnosed with ovarian cancer stages I-IV and collected at time of admission to hospital from awake patients. The collection procedure for U-CAN samples have been described before [36]. In brief, all samples were collected and pre-analytically processed within the same hospital, Akademiska Sjukhuset, Uppsala, Sweden. Samples were transported to the clinical chemistry laboratory where they were automatically processed in room temperature. Samples were centrifuged at 2400× *g* for 8 min. Plasma was then separated and stored at −80 °C on site. Total processing time at the clinical chemistry laboratory from arrival to freezer was 20–30 min. The samples used had not been thawed prior to the analysis described here. The two sets of EDTA plasma samples from the Sahlgrenska Gyncancer Biobank, Göteborg, Sweden, included patients diagnosed with ovarian cancer stages I-IV [38]. The collection procedure has been described before [39]. In brief, the Sahlgrenska patients were recommended over-night fasting, e.g., not eating after midnight the day before surgery and had been administered 500 mL Dextran (ATC: B05AA05, Macrodex^®^ and Promiten^®^). The samples were drawn after general (full) anaesthesia for 10–45 min but prior to surgical incision. Samples were collected and analysed within the same hospital. Samples were stored at 4 °C within 15–30 min and transported to the research laboratory where the plasma was isolated and stored frozen at −80 °C within 30 min of arrival. The samples used had not been thawed prior to the analyses described here. One set of the Sahlgrenska patient samples (*n* = 159) collected at surgery was used as discovery cohort and the second set of samples (*n* = 91) collected at surgery was used as replication cohort. These sets were both compared to the set of samples (*n* = 77) from U-CAN that was collected at time of diagnosis from patient that were awake. Clinical measurements (age, cancer stage and pre-operative Mucin-16 (U/mL)) from one additional cohort from Sahlgrenska consisting of 88 samples with ovarian cancer stages I-IV collected at time of diagnosis from awake patients were also made available to us. Details on this cohort have been published before [18]. All tumours were examined by a pathologist specialized in gynaecologic cancers for histology, grade and stage according to International Federation of Gynecology and Obstetrics (FIGO) standards. Patients that have had received neoadjuvant therapy were excluded from the study. The study was approved by the Regional Ethics Committee in Uppsala (Dnr: 2016/145) and Gothenburg (Dnr: 201-15).

The control cohort consisted of 50 age matched healthy (no reported cancers) women from the Northern Swedish Population Health Study (NSPHS). These controls have been described earlier [37].

### 4.2. Plasma Proteome Characterization

Plasma protein abundance levels were measured using the Proximity Extension Assay (PEA) [40]. The PEA is an affinity-based assay and for each protein, a pair of oligonucleotide-labelled antibody probes bind to the targeted protein and if the two probes are in close proximity, a PCR target sequence is formed by a proximity-dependent DNA polymerization event and the resulting sequence is subsequently detected and quantified using real-time PCR. The resulting abundance levels are given in NPX (Normalized Protein eXpression). The samples were randomized across plates and normalized for any plate effects using the built-in inter-plate controls according to manufacturers’ recommendations. Each proximity extension assay has a lower detection limit for each protein. This is defined as three standard deviation above noise level and is calculated specifically for each assay at run-time based on negative controls that are included in each run. Here, all measurements below this limit were removed from further analysis. Here, the abundance levels of 441 proteins in plasma were analyzed in the discovery cohort and an all-female control cohort by PEA using five Olink Proseek Multiplex panels (Cardiovascular II, Cardiovascular III, Inflammation I, Neurology I and Oncology II) and quantified by real-time PCR using the Fluidigm BioMark™ HD real-time PCR platform [40]. In addition, we have previously characterized the abundance levels of 552 proteins in the discovery and replication cohort using six Proseek Multiplex panels (Cardiometabolic, Cell Regulation, Development, Immune Response, Metabolism and Organ Damage) [18]. Lastly, 41 of the proteins from the 441 proteins and Prolactin was characterized in the replication cohort using dual 21-plex custom panels as described earlier [37,41]. All PEA analyses were carried at Olink Proteomics AB, Uppsala, Sweden. Assay characteristics including assay performance measurements and validations are available from the manufacturer (www.olink.com).

### 4.3. Statistical Analysis

For detection of deviating levels between sedated and awake patients, the protein abundance levels were transformed to z-score by removing the mean and dividing with the standard deviation for each protein. This analysis did not include the control samples. This normalization ensures a mean of 0 and a standard deviation of 1 for each protein. A two-sided *t*-test was used to examine statistical difference in the discovery cohort, the resulting *p*-values were adjusted for multiple hypothesis testing by Bonferroni correction. Any protein with adjusted *q*-value < 0.05 was examined in the replication cohort where the *p*-values were adjusted for the total number of proteins that were brought forward from the discovery phase (Bonferroni correction). In the combined analysis, t-tests were used and adjusted for multiple hypothesis testing as in the discovery stage. Power-calculations were done using the ‘pwr.t2n.test’ function from the ‘pwr’ R-package [42] at 0.05 significance level requiring 95% power. The so-called beanplots in Figure 1b was generated with package ‘beanplot’ [43] (version 1.2) using function ‘beanplot’ with default parameters. The circular diagram in Figure 2a was generated with the ‘circlize’ [44] package (version 0.4.6). Principal component analyses (PCA) in Figure 2b was done with the ’prcomp’ function from the ‘stats’ package (version 3.4.2) with parameters ‘scale.’ and ‘center’ set to ‘TRUE’. All calculations were performed in R [45] (version 3.4.2).

### 4.4. Mouse Human Ortholog Mapping

Mouse genes were mapped to human orthologs through the Ensembl biomart web-resources [46] using the GRCm38.p6 mouse database (Attributes: Gene stable ID, Gene name) and the GRCh38.p12 human database (Attributes: Gene stable ID, HGNC symbol, UniProtKB Gene Name ID).

## 5. Conclusions

We compared abundance levels of up to 983 proteins in a total of 327 patients diagnosed with ovarian cancer. To the best of our knowledge, this is the most comprehensive study to date, both in terms of number of subjects and in the number of proteins, to investigate the effect of pre-operative fasting and general anaesthesia on circulating plasma protein biomarker levels. Our results indicate that over 40% of the plasma proteome is significantly affected by these factors. This implies the need to carefully match cases and controls with respect to pre-analytical procedures immediately before, or during surgery in future protein biomarker discovery studies. Our recommendation is that plasma samples for bio-banking should be collected in awake, non-fasting, patients to minimize bias when comparing with healthy controls, with other study populations or in longitudinal studies with samples at follow-up or recurrence.

## Figures and Tables

**Figure 1 cancers-12-02439-f001:**
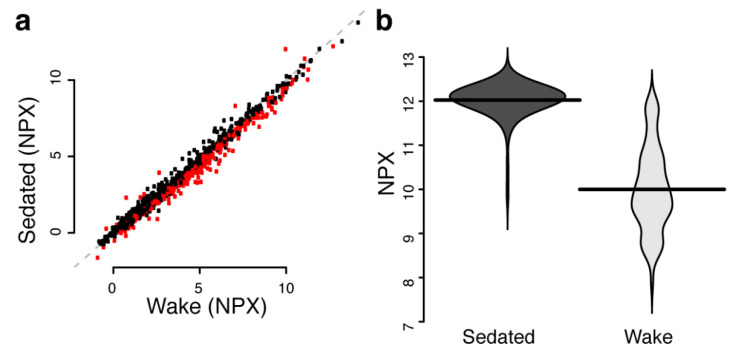
(**a**) Mean protein abundance levels in NPX (Normalized Protein eXpression) in the wake and sedated group. Significant (Bonferroni adjusted *q*-value < 0.05) differences are shown in red. (**b**) Distribution of protein abundance levels (beanplots) in the wake and sedated group for Prolactin. Horizontal lines represent mean values.

**Figure 2 cancers-12-02439-f002:**
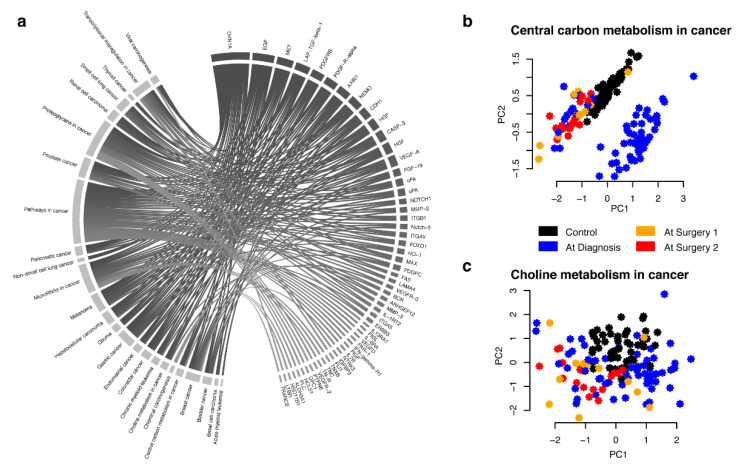
(**a**) KEGG pathways in cancer including genes that code for proteins that have significantly deviating levels between samples collected at diagnosis (awake patients) or at surgery (sedated patients). Pathways are show to the left and proteins to the right. (**b**) Principle component analysis (PCA) of NPX values for proteins with significantly deviating levels between samples collected at diagnosis or at surgery with encoding genes listed in the KEGG pathway “Central carbon metabolism in cancer” (hsa05230). Black dots correspond to healthy female controls, orange and red correspond to samples collected at time of surgery from patients diagnosed with ovarian cancer and blue dots to samples collected at time of diagnosis from patients diagnosed with ovarian cancer. (**c**) as (**b**) but for “Choline metabolism in cancer” (hsa05231).

**Figure 3 cancers-12-02439-f003:**
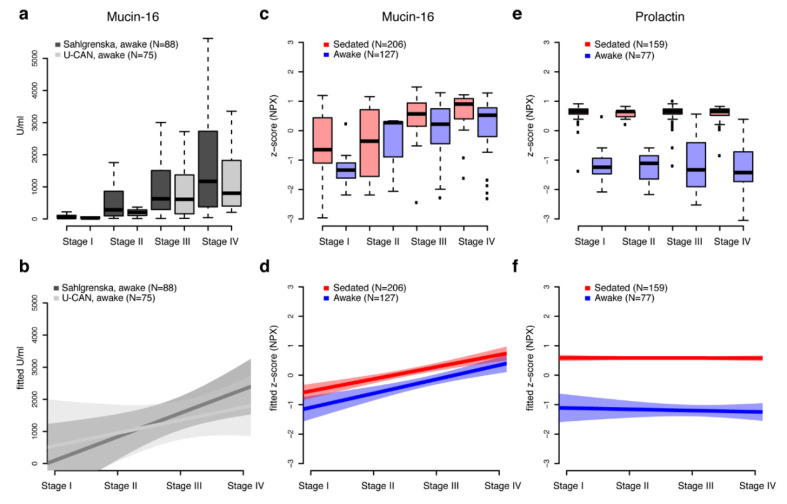
(**a**) Preoperative clinical measurements of Mucin-16 (CA-125) for two sets of awake patient cohorts from the two biobanks, Sahlgrenska (dark grey) and U-CAN (light grey). Measurements are given in U/mL and stratified on ovarian cancer stage (I–IV). Outliers were omitted from the boxplot. The top and the bottom of the box represents the 25th and 75th percentile and the band inside the box the median value. (**b**) Linear regression models (thick lines) and 95% confidence intervals (shaded area) for preoperative measurements of Mucin-16 as function of ovarian cancer stage. (**c**) Comparisons of z-scaled PEA measurements (NPX) of Mucin-16 in sedated and wake patients stratified by ovarian cancer stage. The top and the bottom of the box represents the 25th and 75th percentile and the band inside the box the median value. (**d**) Linear regression models (thick lines) and 95% confidence intervals (shaded area) for z-scaled PEA measurements of Mucin-16 in sedated and wake patients as function of ovarian cancer stage. (**e**,**f**) as (**c**,**d**) but for Prolactin.

**Table 1 cancers-12-02439-t001:** Cohort characteristics

State	Nr.	Sex ^a^	Age (Mean +/sd)	Age Pval ^b^	Panel 1–5 ^c^	42-Plex ^d^	Panel 6–11 ^e^	Ovarian Cancer Stages			
								I	II	III	IV
At Diagnosis	77	1.0	58.8 (12.5)	0.17	50 ^e^	77	77	10	3	40	24
At Surgery 1st	159	1.0	61.8 (11.9)	0.17		159	100	35	9	96	19
At Surgery 2nd	91	1.0	61.8 (12.6)	0.38			79	35	7	33	4
					47			11		36	
Control	50	1.0	58.4 (17.6)	0.39	50						

^a^ Fraction of women in each cohort. ^b^ Lowest nominal *p*-value (two-sided Wilcoxon-text) for age-distribution in this cohort compared with any of the other cohorts. ^c^ Cardiovascular II, Cardiovascular III, Inflammation, Neurology and Oncology. This data is referred to as Appendix A, “At Diagnosis” and “At surgery 2” only. ^d^ 41 proteins from Panel 1–5 ^c^ plus Prolactin. This data is referred to as Appendix A. ^e^ Cardiometabolic, Cell Regulation, Development, Immune Response, Metabolism and Organ Damage [18] This data is referred to as Appendix A.

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
