# Peer review of "Preoperative Fasting and General Anaesthesia Alter the Plasma Proteome"

_cancers, 2020, doi:10.3390/cancers12092439_

Round 1

Reviewer 1 Report

This manuscript describes the use of proximity extension assays (PEAs) using multiple Olink panels to monitor the role of preoperative fasting and general anaesthesia on the plasma proteome. The manuscript is generally well written and informative. Comprehensive data sets have been supplied. The results are well presented and discussed, and caveats noted. The take home message that there is a need for careful matching of the pre-analytical conditions of samples collected from controls to cases at time of surgery in the discovery and clinical use of protein biomarkers is important. The following points should be addressed.

How were variables such as circadian rhythm, hormonal variation (menstruation, menopause), co-morbidities etc. taken into account?

Line 85. It is not clear whether the same patients were analysed both when awake and post anesthesia. Please clarify.

Line 90. … implications …

Author Response

This manuscript describes the use of proximity extension assays (PEAs) using multiple Olink panels to monitor the role of preoperative fasting and general anaesthesia on the plasma proteome. The manuscript is generally well written and informative. Comprehensive data sets have been supplied. The results are well presented and discussed, and caveats noted. The take home message that there is a need for careful matching of the pre-analytical conditions of samples collected from controls to cases at time of surgery in the discovery and clinical use of protein biomarkers is important. The following points should be addressed.

How were variables such as circadian rhythm, hormonal variation (menstruation, menopause), co-morbidities etc. taken into account?

Author reply: Unfortunately, none of these variables were explicitly recorded for the samples used here. Assumptions on menopausal status could perhaps be made from individual age but was not done here. The data was analyzed on group level, and the age-distribution was not statistically different between the analyzed groups (all p-values > 0.17, Table 1). There was also no difference in the fraction of proteins that are known to be affected by e.g. lifestyle variables and use of common medications among the proteins that we found to be affected by sedation/fasting and those that were not found to be affected here. We do however agree with the reviewer that the aforementioned variables are important and have added a section on the lack of these in our study in the discussion that reads:

Neither did we have access to detailed records of co-morbidity, menopausal stage, circadian rhythm or lifestyle choices such as smoking status or alcohol use. However, we analysed all stages and ages jointly and in our largest material, including the data for protein panels 6-11 (Table 1), over 85% of the associations found in the discovery step did replicate, suggesting that the results are robust despite the actual differences in stage-distribution and possible other differences between the cohorts.” Lines 257-262 in the revised manuscript.

Line 85. It is not clear whether the same patients were analysed both when awake and post anesthesia. Please clarify.

Author reply: We acknowledge the unclarity of this in this paragraph and have added the following sentence to clarify “No patients overlapped between the three cohorts.” Lines 86-87 in the revised manuscript.

Line 90. … implications …

Author reply: The sentence has been changed and no read “Finally, we demonstrate how these effects can yield false positive results by comparing the plasma protein levels in the cases to awake controls with implications in cancer-related pathways.” Lines 89-91 in the revised manuscript.

Reviewer 2 Report

This is a well-conceived and well-written study showing yet another concerning variable in protein biomarker studies in human plasma, using an increasingly interesting proteomic system (Olink). 

Major comment/concerns:

There is a common theme in this work in that I think it would be difficult for someone else to take this study and the data presented herein and reproduce the results. This can be easily rectified with by providing 1) a thorough key to describe the data points in the supplemental and by 2) describing how the figures were generated. 

1) Unless I'm missing a table, I don't see how to define what sample belongs to what cohort and what measurement is derived from which assay. The supplemental obviously contains a lot of data, but I don't see how another group could make use of this work without a key.

I would suggest an additional supplemental table be added to the manuscript (or a larger, more informative version of Table 1) to facilitate another group to repeat your steps or to reanalyze your data with a different set of tools. 

2) The figures are great, but they need documentation. It's easy to think "it's just a PCA plot" but with 10,000 different packages available on CRAN, documentation is becoming more critical each day. The RMarkdown, or Jupiter, etc., would be best but adding how each chart was made, with what package and settings would be sufficient. 

Minor notes: 

I do think that the recent study from Jeffs et al., that attempts to quantify chemical effects of freeze/thaw cycles would be a useful reference in the text. (https://doi.org/10.1074/mcp.TIR119.001659). Not critical, but it seemed like the reference that fit there. It just came to mind as the reference that I would put on that line. 

I appreciate the level of detail that went into the planning the sample prep. Given these cohorts, their staggering size and their origin, I don't know how you could have planned this study better. Bravo for setting this up the hard (right) way.

Author Response

This is a well-conceived and well-written study showing yet another concerning variable in protein biomarker studies in human plasma, using an increasingly interesting proteomic system (Olink). 

Major comment/concerns:

There is a common theme in this work in that I think it would be difficult for someone else to take this study and the data presented herein and reproduce the results. This can be easily rectified with by providing 1) a thorough key to describe the data points in the supplemental and by 2) describing how the figures were generated. 

Author reply:  We agree, please find detailed replies to both issues below.

1) Unless I'm missing a table, I don't see how to define what sample belongs to what cohort and what measurement is derived from which assay. The supplemental obviously contains a lot of data, but I don't see how another group could make use of this work without a key.

I would suggest an additional supplemental table be added to the manuscript (or a larger, more informative version of Table 1) to facilitate another group to repeat your steps or to reanalyze your data with a different set of tools. 

Author reply: Although this seemed clear to us at the time of manuscript compilation, we now completely agree with the reviewer that additional information is needed. We have added descriptions in the supplementary tables explaining which data set that belongs to which cohort. We also added cross-references in the captions for Table 1 which now reads:

a Fraction of women in each cohort. b Lowest nominal p-value (two-sided Wilcoxon-text) for age-distribution in this cohort compared with any of the other cohorts. c Cardiovascular II, Cardiovascular III, Inflammation, Neurology and Oncology. This data is referred to as SData1 in the Supplementary material, “At Diagnosis” and “At surgery 2” only. d 41 proteins from Panel 1-5c plus Prolactin. This data is referred to as SData2 in the Supplementary material. e Cardiometabolic, Cell Regulation, Development, Immune Response, Metabolism and Organ Damage[18] This data is referred to as SData3 in the Supplementary material.” Lines 119-124 in the revised manuscript.

2) The figures are great, but they need documentation. It's easy to think "it's just a PCA plot" but with 10,000 different packages available on CRAN, documentation is becoming more critical each day. The RMarkdown, or Jupiter, etc., would be best but adding how each chart was made, with what package and settings would be sufficient. 

Author reply: We thank the reviewer for pointing this out. We agree that this is often overlooked and adding details is important. To clarify how the non-standard figures (e.g. apart from boxplots and scatterplots) were created we have added the following to the methods section:

The so-called beanplots in Figure 1b was generated with package ‘beanplot’[44] (version 1.2) using function ‘beanplot’ with default parameters. The circular diagram in Figure 2a was generated with the ‘circlize’[45] package (version 0.4.6). Principal component analyses (PCA) in Figure 2b was done with the ’prcomp’ function from the ‘stats’ package (version 3.4.2) with parameters ‘scale.’ and ‘center’ set to ‘TRUE’.” Lines 404-408 in the revised manuscript.

Minor notes: 

I do think that the recent study from Jeffs et al., that attempts to quantify chemical effects of freeze/thaw cycles would be a useful reference in the text. (https://doi.org/10.1074/mcp.TIR119.001659). Not critical, but it seemed like the reference that fit there. It just came to mind as the reference that I would put on that line. 

Author reply: The mentioned study is interesting and provides a very important contribution into quantifying measurable effects of pre-analytical variation, namely freeze/thaw cycles. As all samples used here have undergone the same number (1) of freeze/thaw cycles prior to analysis, the contributed variance of this is as small as it can be and we think the mentioned reference is slightly off-focus for the current manuscript and decided to leave it out.

I appreciate the level of detail that went into the planning the sample prep. Given these cohorts, their staggering size and their origin, I don't know how you could have planned this study better. Bravo for setting this up the hard (right) way.

Author reply:  Thank you.